# Temporal dynamic vulnerability - Impact of antecedent events on residential building losses to wind storm events in Germany

Andreas Trojand[1], Henning W. Rust[1,2], and Uwe Ulbrich[1]

[1]Institute for Meteorology, Freie Universität Berlin, Carl-Heinrich-Becker Weg 6-10, 12165 Berlin
[2]Hans-Ertel-Centre for Weather Research, Carl-Heinrich-Becker Weg 6-10, 12165 Berlin

**Correspondence:** Andreas Trojand (andreas.trojand@fu-berlin.de)

**Abstract.**

Severe winter storm events are one of Central Europe's most damaging natural hazards and are therefore particular in focus for disaster risk management. One key factor for risk is vulnerability. Risk assessments often assume vulnerability to be constant. This is, however, not always a justifiable assumption. This work seeks and quantifies a potential dynamic of vulnerability for residential buildings in Germany. A likely factor affecting the dynamics of vulnerability is the hazard itself (Aerts et al., 2018). As an extreme events may destroy the most vulnerable elements, it is likely that the subsequent rebuilding or repair will reduce their vulnerability for following events (UNISDR, 2017). Therefore, the intensity of the previous events and the resulting damage can be assumed to be a decisive factor in changing vulnerability. A second important factor is the time period between the previous and current event. If the next event occurs during the reconstruction phase, vulnerability might be higher than when the reconstruction phase is completed (de Ruiter et al., 2020).

Here, we analyze the importance of previous storm events for the vulnerability of residential buildings. For this purpose, generalized additive models are implemented to estimate vulnerability as a function of the intensity of the previous event and the time interval between the events. The damage is extracted from a 23-year-long data set of the daily storm and hail losses for insured residential buildings in Germany on the administrative district level provided by the German Insurance Association, and the hazard component is described by the daily maximum wind load calculated from the ERA5 reanalysis. The results show a negative relationship between the previous event's intensity and the current event's damage. As the time since the previous event increases, a significant decrease in an event's associated damage is found. On a daily scale, the first five to ten days are especially crucial for vulnerability reduction.

## 1  Introduction

Severe wind storm events resulting from extratropical cyclones significantly impact economic losses in Central Europe. Although impact on human life is relatively small and the damage to individual buildings by wind storms is generally moderate, the accumulated damage from 2002 to 2021 due to storm events caused three times more damage to residential buildings than other natural hazards (MunichRe, 2023; GDV, 2023). The main reason for this is the frequency of wind storm events coupled with spatial expansion that leads to a significant number of insurance claims and results in high total losses (Sparks et al.,

1994; MunichRe, 2023). Therefore, risk assessments play a crucial role in analysing recent and predicting those of future occurrences. Here, we define risk as a function of hazard, exposure, and vulnerability (e.g. UNDRO, 1980; UNDRR, 2021; IPCC, 2022). While all three components are essential for understanding risk, this study places particular emphasis on vulnerability due to its complex and multifaceted nature. Accordingly, we adopt the following definition of vulnerability: The condition determined by physical, social, economic and environmental factors or processes that increase the susceptibility of an individual,

a community, assets, or systems to the impacts of hazards (UNDRR, 2021).

Extensive research has been conducted on storm events in Western European countries due to their significant impact, with most studies focusing on the hazard component of risk assessments and not on the vulnerability. Klawa and Ulbrich (2003) developed a storm loss index based on the assumption that the loss increases with the cube of normalized gust intensity in excess of the 98th percentile threshold, which was adapted by, amongst others, Pardowitz et al. (2016) for probabilistic

prediction of wind storm damage. Röösli et al. (2021) utilized ensemble weather predictions to forecast winter storm impacts in Switzerland. The spatial scale of damage models ranges from regional (Donat et al., 2010) to federal states (Heneka et al., 2006) to Europe-wide (Koks and Haer, 2020). In several publications, return periods have been calculated (e.g. Heneka and Hofherr, 2011; Donat et al., 2011a) and expected future changes due to climate change have been investigated (e.g. Dorland et al., 1999; Schwierz et al., 2009; Donat et al., 2011b). A typical approach, also used in some of the above studies (e.g. Klawa

and Ulbrich, 2003; Heneka et al., 2006; Donat et al., 2010), is to relate the gust speed to a local gust percentile; this is a way to include spatial variability of vulnerability. However, all of these studies assume vulnerability to be constant over time. This has been criticized by, amongst others, by Aerts et al. (2018) and Cremen et al. (2022).

Understanding vulnerability, including its physical, social, economic and environmental factors and their potential changes, is crucial for improving risk assessment (Formetta and Feyen, 2019). All these conditions change over time, and thereby, so does

vulnerability changes over time. Assuming stationary vulnerability may lead to over- or underestimation of risk (Aerts et al., 2018; de Ruiter and van Loon, 2022). After an event, existing risk assessments rapidly become outdated as the vulnerability changes due to the event itself (Gill and Malamud, 2016). Therefore, many studies advocate for a dynamic approach (e.g. Papathoma-Köhle et al., 2012; Di Baldassarre et al., 2018). This study aims to detect and quantify the temporal dynamics of the physical vulnerability of residential buildings in Germany due to wind storm events. Physical vulnerability is derived from

vulnerability curves that describe the relationship between wind storm intensity and the resulting damage.

In disaster risk management, temporal dynamic vulnerabilities can be categorized into two groups. The first group refers to the underlying dynamics such as an increase in gross national product, technical progress, long-term deterioration of buildings (Stewart et al., 2011, 2012) or lack of maintenance (Orlandini et al., 2015). These general changes also arise even if no hazard occurs and therefore can be described as non-hazard-specific dynamics (de Ruiter and van Loon, 2022; Fuchs and Glade,

2016). Although Simpson et al. (2021) and Drakes and Tate (2022) discuss hazard dynamics for social vulnerability, this has not been included – to our knowledge – in physical vulnerability assessments so far. The second group involves dynamics as a consequence of the hazard itself. This type of dynamic vulnerability can be further divided into short-term and long-term effects. In the long term, the idea of "build back better" (UNISDR, 2017), which refers to the recovery phase in the aftermath of an event, is a key factor. During this phase, there is the opportunity not just to restore the status before the event, but to reduce

vulnerability by improving the construction. Nikkanen et al. (2021) found that people who suffered from storm impacts in the past were more likely to prepare and therefore reduced the risk. After very severe events, there have been cases of an imposed change in building standards (Stewart, 2003; Walker, 2011; Stewart, 2013). Stewart and Li (2010) investigated changes in vulnerability due to a new home building code for Queensland, Australia, showing a decrease in vulnerability due to improved building standards. Case studies show a reduction in vulnerability due to previous events (from here: pre-events) for various types of hazards (e.g. Kreibich et al., 2017; Becker et al., 2017; Kreibich et al., 2023). However, these studies do not include the intensity of pre-events, although it is likely to affect the reduction in physical vulnerability substantially. The assumption is that the most vulnerable buildings get damaged and, consequently, reconstructed more stably and are thus less vulnerable to the next event. Therefore, the first research objective of this study is to quantify the effect of pre-events' intensity on vulnerability dynamics.

In the short term, important factors for dynamics in vulnerability are the occurrence of consecutive and compound events (Clark-Ginsberg et al., 2018). The latter is outside of the scope of this study as we are restricting ourselves to the effect of one type of hazard. Consecutive events can cause more significant damage than isolated events (Marzocchi et al., 2012; Gill and Malamud, 2014; Aerts et al., 2018; de Ruiter et al., 2020), as the time between two events can substantially change the vulnerability to the second event (Gill and Malamud, 2014; de Ruiter et al., 2020). The time between two events is crucial for winter storm events in Germany, as they occur over a short period. For example, the series of wind storms Ylenia, Zeynap and Antonia occurred within a few days in February 2022 (Mühr et al., 2022). With substantial time between two events, the vulnerability might decrease due to preparedness (Gill and Malamud, 2016). Rathfon et al. (2012) point out that the speed of post-event housing reconstruction has not yet been quantitatively described and that, since said reconstruction is crucial to the overall recovery of the community, an in-depth analysis is necessary. Therefore, the second research objective of this study is to quantify the impact of the time between two events on the dynamics of vulnerability.

Although many studies deal with risk assessments of winter storms in Europe, a comprehensive study of how vulnerability changes over time is still lacking. This study aims to fill part of this gap by including the influence of the intensity of pre-events as well as the time between two events in the risk analysis. This allows for more detailed analyses of the temporal dynamics of vulnerability.

The data sets used for the analysis are described in section 2. In the subsequent section on methods (section 3), the calculation of the hazard component is first described (subsection 3.1), followed by the definition of *events* and *pre-events* (subsection 3.2) and then the spatial allocation of meteorological and insurance data (subsection 3.3). Subsection 3.4 provides a theoretical background to generalized additive models before we explain the model setup in subsection 3.5. The results of different models are shown in section 4 and discussed in section 5, followed by the conclusion in section 6.

## 2    Data

Two data sets are used to quantify the temporal dynamic vulnerability of residential buildings in Germany due to wind storm events. The first data set, provided by the German Insurance Association (Gesamtverband der deutschen Versicherungswirtschaft

- GDV), contains information about the loss and exposure of residential buildings. The ERA5 data from the European Centre for Medium-Range Weather Forecasts (ECMWF) is used for the meteorological part of the analysis.

## 2.1 Insurance Data

The GDV provided a 23-year data record on insured losses for residential buildings in Germany. These records contain losses, the number of claims on a daily basis, the insured sum and the number of contracts accumulated at the administrative district level from 1997 to 2019. The data set comprises only losses from storm and hail events. However, it is not possible to distinguish between the two causes for the damage from the data. As hail typically results from summer thunderstorms, we focus on the winter half-year, spanning October through March, to exclude damage from hail. Furthermore we define the loss ratio as

$$\text{loss ratio} = \frac{\text{total loss in €}}{\text{insured sum in €}}. \tag{1}$$

Using the loss ratio, and not the total loss, has three advantages: first, with the total insured sum, the exposure and its temporal changes are included in the modelling approach. Second, the division by the insured sum adjusts for inflation. Lastly, with standardization, administrative districts, which have different sizes ranging from approximately $35\text{km}^2$ for urban municipalities ("Kreisefreie Städte") up to $4500\text{km}^2$ for rural districts ("Landkreise") and different building densities are comparable.

## 2.2 Meteorological Data

For the hazard component of the model, the ERA5 reanalysis data, produced by the ECMWF, are used. This reanalysis is based on 4D-data assimilation and the model forecasts in the CY41R2 Integrated Forcast System (ECMWF, 2016). The ERA5 data has a gridded spatial resolution of $31\text{km}$ for the hourly realization of analysis and short forecasts (18 hours).

## 3 Methods

Based on the data set described in section 2, we first derive the hazard component for the model from the reanalysis (subsection 3.1). In the next step, events and pre-events are defined (subsection 3.2). For the assignment of a meteorological to the insurance data, a spatial allocation is needed, which is explained in subsection 3.3. The theoretical basics of generalized additive models is shown in subsection 3.4, before the model setup with the different covariates is explained subsection 3.5.

## 3.1 Hazard component from reanalysis

Most studies on wind storm impacts use the maximum *wind gust* $(m/s)$ (e.g. Dorland et al., 1999; Heneka and Hofherr, 2011; Pardowitz, 2015) as the hazard component. In this study, we use the daily maximum *wind load* $(\text{N/m}^2)$

$$q = \frac{\rho}{2} v_{\text{gust}}^2, \tag{2}$$

with $v_{\text{gust}}$ (m/s) representing the maximum daily wind gust and $\rho$ the air density $(\text{kg/m}^3)$ at the hour of the daily maximum wind gust. As air density is not provided with the ERA5 data, we calculate it as

$$\rho = \frac{p}{R_s T},\tag{3}$$

with $p$ the surface pressure (hPa), $R_S$ the specific gas constant (287.052 J/(kgK)) and $T$ the temperature (K). The conversion from wind speed to wind load under physical standard conditions for some relevant classes of the Beaufort scale is shown in table 1.

**Table 1.** Wind load according to the Beaufort scale (WMO, 1970) and corresponding wind speed under physical standard conditions of 0°C and 1013.25hPa air pressure.

| Beaufort | Description | Wind speed | Wind load |
|:---:|---|---|---|
| 8 | Gale | 17.2 - 20.7m/s | 191 - 278N/m$^2$ |
| 9 | Strong gale | 20.8 - 24.4m/s | 279 - 386N/m$^2$ |
| 10 | Storm | 24.5 - 28.4m/s | 387 - 523N/m$^2$ |
| 11 | Violent storm | 28.5 - 32.6m/s | 524 - 688N/m$^2$ |
| 12 | Hurricane force | $\geq 32.7$m/s | $\geq 689$N/m$^2$ |

Wind load is chosen over wind gust for two reasons: first, Minola et al. (2020) show higher biases of ERA5 wind speed and wind gust for mountainous regions compared to inland or coastal areas. With coastal areas in the north and mountainous regions in the south of Germany, we need to consider these differences in biases. Wind load takes into account air pressure, which can be used as an approximation for elevation, based on the barometric formula. The second reason is that building codes use wind load for construction standards. Germany is divided into four wind load zones with different standards based
on European and German norms (ENV-1991-24, 1995; DIN 1055, 2005). Although these zones are calculated using wind speed, for construction purposes, the transfer to wind load with units N/m$^2$ is more appropriate than wind speed in m/s.

### 3.2 Definition of Event and Pre-Event

We follow the rules of homeowner insurance in Germany (Wohngebäudeversicherung (GDV, 2022)): an event is defined by wind speeds of at least 17.2m/s (Beaufort scale 8 (WMO, 1970)). For this study, we used this threshold for the daily maximum
ERA5 wind gust. If consecutive days exceed this threshold, these days are considered as belonging to one event with the maximum wind load of these days assigned to it; the associated damage is accumulated over the event days. As a consequence, there is always at least one full day between events in which the threshold value is not exceeded. Figure 1 shows the distribution of events with an exponential increase in the loss ratio with increasing wind load.

    The definition of a pre-events is different: it is assumed that minor damage will not have a significant impact on the entire
administrative district for the next event, as the loss ratio is accumulated over the entire area. Therefore, a threshold for the loss

ratio is used instead of a threshold for the wind gust. For a pre-event we require the loss ratio to exceed 0.01‰ (Figure 1 - red line) in the same administrative district as the occurrence of the event. The choice of 0.01‰ is based on insurance definitions for different event intensities. One threshold is defined as the mean accumulated insurance claims for one month. If this threshold is exceeded by an event on one day, it is a noticeable event (GDV, 2023). We transferred this approach from insurance claims
to the loss ratio and calculated the mean accumulated loss ratio per month, which is 0.0104‰.

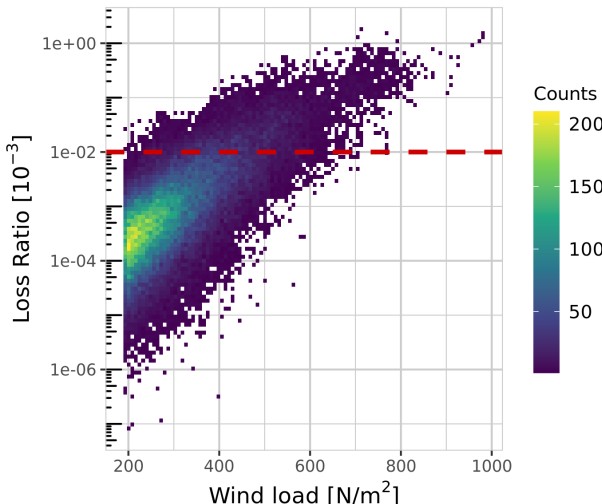

**Figure 1.** Counts of events with wind loads and the related loss ratio on logarithmic scale. Each event is one wind storm in one district with the corresponding loss ratio in the same district. Threshold for pre-events (dashed red line) = 0.01‰.

With the definitions of events and pre-events, each event is paired with the nearest pre-event in time and the days between pre-event and event is calculated.

### 3.3 Spatial allocation of meteorological and insurance data

As the meteorological data are provided on a regular grid and the insurance data at the district level, proper spatial allocation
of the two data sets must be carried out. Previous studies, such as Donat et al. (2011a) and Pardowitz et al. (2016), assigned the district midpoints to the nearest grid point of the meteorological data. In the case of ERA5 data with a spatial resolution of around 31km, this leads to the assignment of several districts to the same grid point in every time step (Figure A1). Therefore, we assign all ERA5 grid points within a 31km buffer to each district (Figure A2). For each time step (event), only the strongest wind load of the allocated grid points is selected and assigned to the event. In addition, taking into account more than one grid
point, especially for larger districts, the probability of missing high wind loads decreases. We choose the buffer of 31km (the same as the ERA5 grid), since we can therefore guarantee that at least four grid points are assigned to each district. On the one hand, theoretically, even a very small district would be allocated to four grid points with the choice of 31km, on the other hand, this buffer is small enough to not include grid points which have no influence on the district and possible damages (Figure A3).

## 3.4 Generalized additive models

The expected loss ratio $\mathbb{E}[LR]$ for a certain event can be described with a generalized *linear* model

$$\mathbb{E}[LR] = g^{-1}\left(\boldsymbol{X_i}\boldsymbol{\beta}\right) \tag{4}$$

with inverse link function $g^{-1}(.)$, $j$ covariates $\boldsymbol{X_i} = (X_{i1}, ..., (X_{ij})$, where $\boldsymbol{\beta} = (\beta_1, ..., \beta_j)$ are the corresponding model parameters to be estimated.

Several authors (e.g. de Jong and Heller, 2008; Laudagé et al., 2019; Garrido et al., 2016) use a Gamma distributed random variable with a logarithmic link function $g(x) = \log(x)$, as the variance of the observed loss ratio increases with the expected value. Here, we use generalized *additive* models to allow more flexibility than in the case of generalized *linear* models; the latter is a typical choice for damage models (e.g. Donat et al., 2011b; Pardowitz et al., 2016). Generalized additive models (Hastie and Tibshirani, 1986; Wood, 2017) are an extension of generalized linear models using smooth instead of linear functions of covariates. This leads to

$$\mathbb{E}[LR] = g^{-1}\left(\beta_0 + \sum_{j=1}^{J} f_j(x_j)\right), \tag{5}$$

where $f_i$ are the smooth functions of the covariate $x_j$ and $\beta_0$ is the intercept.

Several choices are possible for the smoothers (Wood, 2017); here, we use an extension of cubic regression splines with adjusted shrinkage smoothing parameters. Cubic regression splines are composed of piecewise cubic polynomials over the data interval. The intervals are defined by knots and the knot locations are evenly spaced along the covariates. The parameters of cubic regression splines can never all be estimated to be zero and thus an influence might be attributed to the covariate that it does not actually have. We therefore use the extension to cubic regression splines with shrinkage. With this addition, covariates that do not have an influence on the response variable can be eliminated (Wood, 2017). The optimum wiggliness for each smoothing results in fitting the data well, but not over-fitting. The hyper-parameter $\lambda$ controls the wiggliness of the smoothing terms and can be determined by different approaches. Here, we use *Restricted Maximum Likelihood* (REML). All modelling processes and analyses were carried out using `R Statistical Software` (v4.1.2; R Core Team (2021)) and the package `mgcv` (Wood, 2017).

One important aspect in our study is interactions. We use the interaction between the pre-event loss ratio and the event's wind gust, as well as between pre-event loss ratio and the time between the events for describing the loss ratio. For the implementation of the interaction between two or more continuous covariates that vary on different scales, tensor products are used. In case of an interaction between a continuous covariate and a categorical covariate, a factor-smooth interaction is suggested for this case by Wood (2017).

## 3.5 Model Setup

Past studies include only a hazard component and exposure parameters as covariates in the approach to damage models (e.g. Heneka et al., 2006; Pardowitz et al., 2016; Welker et al., 2021). To quantify the temporal dynamics of vulnerability due to

pre-events, additional predictors are implemented to account for these changes. The dynamics of vulnerability can be quantified by analyzing the influence of additional covariates on the hazard's impact on the loss ratio.

    In total, four different models are developed (Table 2). In all four models the event's loss ratio serves as the response variable and the event's wind load as a covariate for the hazard. In model $M_{preEvent}$ only the loss ratio of the pre-event within the same administrative district is implemented so that the effect of the pre-events in general without taking the time between two events

into account, can be quantified.

    The three models $M_{Season}$, $M_{Weeks}$, and $M_{Days}$ also include time as a covariate distinguishing a seasonal, weekly, and diurnal time scales. Model $M_{Season}$ uses the categorical variable *season between* for the time between events. For event – pre-event combinations labelled *same-season* the pre-event occurred within the same season as the event, which means that at least one event with a loss ratio larger than 0.01‰ happened before the event we are looking at, but still within the same winter

half-year. If several pre-events occur within the same winter half, only the immediately preceding pre-event is assigned as the pre-event to the event. If there is no same-season pre-event, the most damaging event of the previous season is assigned to the event and the combination is labelled as *pre-season*.

    For model $M_{Weeks}$ we reduce the data set and only use pairs of events and pre-events occurring within the same season. For each pair of pre-event and event the weeks between is calculated and used as the covariate *weeks between*. If the pre-event and

event occurred within seven days there are *zero weeks* between the events and in the case that the pre-event occurs in the first week of the winter season and the event occurs in the last week , there are *25 weeks* between the events. For the model $M_{Weeks}$ we only take pre-event – event combinations into account with a maximum of 14 weeks in between. These make up 95% of all data within one season, reduces the influence of observed pre-event – event outliers and still includes the most important time range of greater than three months after an event to recover.

In model $M_{Days}$ the input data is reduced one more time. For this model only event – pre-event combinations are implemented where the time between the pre-event and the event is at most 28 days (4 weeks). These three different time scales are chosen, as we expect the daily scale to be essential shortly after the event. In this time the duration of the reconstruction phase is crucial (Rathfon et al., 2012). In addition, during the reconstruction phase an increase in vulnerability is likely (Kappes et al., 2012; Goebel et al., 2015), which can be analysed by using daily scale more adequately than by using the other two models.

However, on longer timescales it is more appropriate to group the time between events into weeks or seasons. Using a daily time scale poses difficulties for the modeling framework when the pre-event and event periods are separated by more than one summer season, as the temporal gap between two winter seasons disrupts continuity of days between events.

    The additional covariate mean value 1914 ($mV_{1914}$) is included to describe the general development of buildings within a district, to ensure that the model results for the covariate describing the intensity of the pre-event do not include general trends

in vulnerability. $mV_{1914}$ is based on the value 1914 ("Wert 1914") (GDV, 2024) and the sum of contracts in a district. Details on the value 1914 and the results are described in Appendix B.

**Table 2.** Covariates used in the four different generalized additive models

| Model name | Hazard | Vulnerability | | Baseline |
| --- | --- | --- | --- | --- |
| | | Pre-Event | Time Between Events | |
| $M_{preEvent}$ | Wind Load | Pre-Event Loss Ratio | - | $mV_{1914}$ |
| $M_{Season}$ | " | " | Season Between | " |
| $M_{Weeks}$ | " | " | Weeks Between | " |
| $M_{Days}$ | " | " | Days Between | " |

## 4   Results

We identified 70703 events in the 23-year data record for the 401 administrative district in Germany based on the definitions described in subsection 3.2. Almost 60% of all events have a wind load lower than $300\mathrm{N/m^2}$ (Beaufort scale 8) and nearly 40% of all pre-event loss ratios are lower than 0.02‰ (Figure 2). The number of events decreases with higher wind loads and more damaging pre-events.

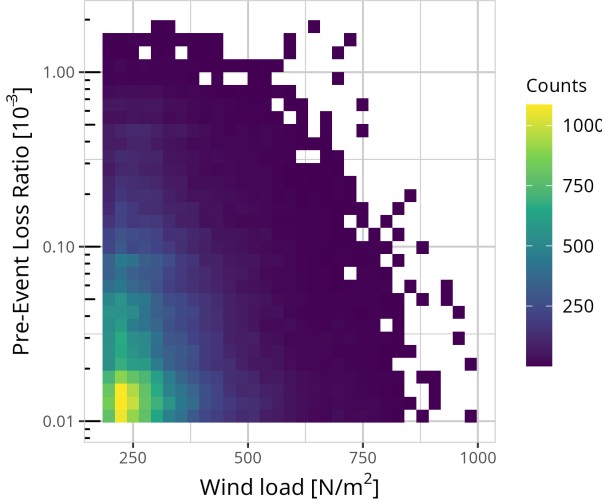

**Figure 2.** Counts of events with wind loads and the related pre-event loss ratio.

### 4.1   Intensity of pre-events

First, we compute the effect of pre-event loss ratios on the vulnerability without taking the time between two events into account. Figure 3 shows the results of the model $M_{preEvent}$ for three different fixed event intensities (wind load). The predom-

inant factor influencing the loss ratio is the wind load. With higher wind load the loss ratio increases (see three different curves in Figure 3). The expected loss ratio for an event depends not only on the wind load but also on the loss ratio associated with the pre-event (abscissa in Figure 3). For a given wind load, the expected loss ratio for the event decreases with increasing loss ratio of the pre-event. Thus, the vulnerability is lower in cases with higher pre-event loss ratios.

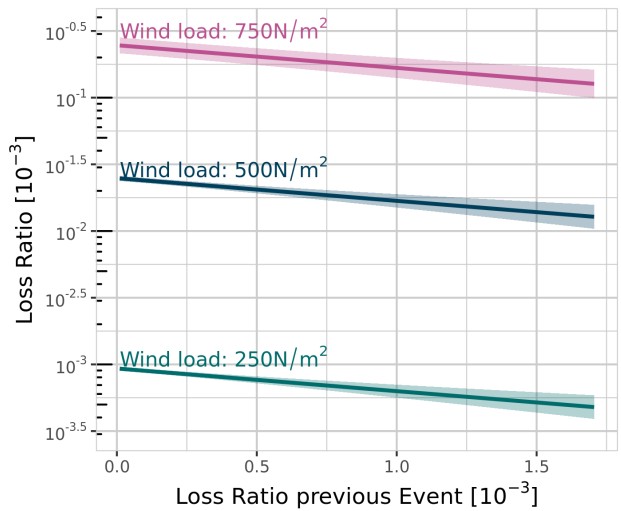

**Figure 3.** Results of Model $M_{preEvent}$ showing the relationship between loss ratio of an event and the loss ratio of the pre-event, exemplarily for three different wind loads, with their 95% confidence interval (shaded areas).

Using the uncertainty estimates from Model $M_{preEvent}$, we can obtain a minimum pre-event loss ratio that marks a statis-
tically significant change in the event's loss ratio from a value obtained for a pre-event loss ratio of 0.01‰. Based on a $t$-test and three significance levels ($\alpha \in \{0.1, 0.05, 0.01\}$), Table 3 shows these critical values exemplarily for the three wind loads shown in Figure 3.

**Table 3.** Critical values for pre-event loss ratio for three significance levels of a two-sided $t$-test given exemplarily for the three wind loads shown in Figure 3. If the pre-event loss ratio exceeds the value given in the table, the associated event's loss ratio is significantly different from the loss ratio expected for a pre-event loss ratio of 0.01‰.

| Event intensity | Significance level $\alpha$ | | |
|---|---|---|---|
| | 10% | 5% | 1% |
| 250m/s | 0.17‰ | 0.21‰ | 0.28‰ |
| 500m/s | 0.21‰ | 0.25‰ | 0.34‰ |
| 750m/s | 0.62‰ | 0.76‰ | 1.07‰ |

## 4.2 Temporal Impacts

### 4.2.1 Seasons

21.639 pre-events occurred within the same winter half-year as the event itself, while 49.064 pre-events happened at least one winter season before the event.

To quantify the effect of time between two events on vulnerability dynamics, Figure 4 shows the vulnerability curves for the same-season and pre-season pre-events with two different fixed pre-event loss ratios.

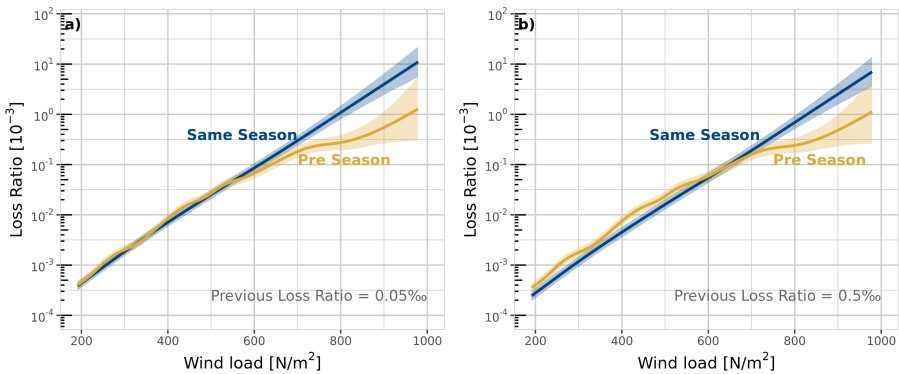

**Figure 4.** Comparison of events happening with same-season pre-events (blue) or with pre-season pre-events (yellow) for a fixed pre-event of **a)** 0.05‰ and **b)** 0.5‰. Shaded area the 95% confidence interval.

For an event with a pre-event loss ratio of 0.05‰, wind loads higher than about $550\mathrm{N/m^2}$ (Beaufort Scale 11 - "violent
storm") lead to a statistically significant reduction in vulnerability for events with a pre-season pre-event compared to events with a same-season pre-event. This significant difference for a pre-event loss ratio of 0.5‰ begins at $650\mathrm{N/m^2}$. With increasing wind load, both the differences in vulnerability and the uncertainties of the model results increases.

There is no clear trend for wind loads below $550\mathrm{N/m^2}$ for pre-event loss ratios of 0.05‰. For pre-event loss ratios of 0.5‰ and wind loads lower than $650\mathrm{N/m^2}$, the vulnerability is greater for events with pre-season pre-events compared to those with
same-season pre-events.

A detailed analysis of the differences for the entire range of pre-event loss ratios is shown in Figure 5. For events with a wind load lower than $550\mathrm{N/m^2}$ the loss ratio is in general higher for events with pre-events occurring at least one winter season before than for events happening within the same season as the pre-event.

Events between $500\mathrm{N/m^2}$ and $700\mathrm{N/m^2}$ (12 Beaufort) with pre-events loss ratios greater than 0.5‰ also show lower loss
ratios for cases where event and pre-event occurred within the same winter season rather than with one or more seasons in between. With decreasing pre-event loss ratios, the events' loss ratio is higher for same-season pre-events compared to pre-season pre-events.

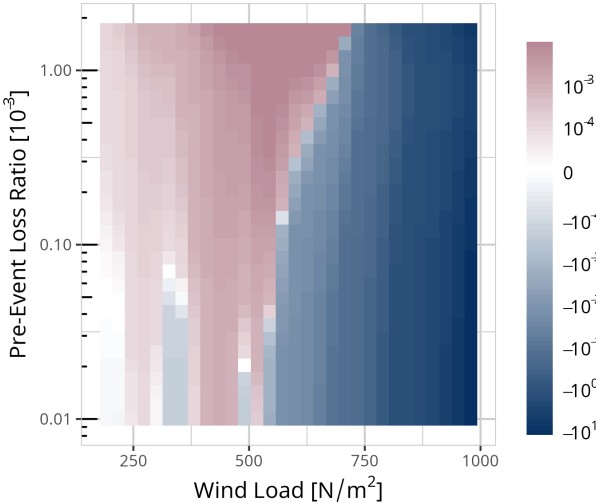

**Figure 5.** Absolute loss ratio difference between events with same season pre-events and events with pre-season pre-events. Blue colours indicate a higher loss ratio for same season events, red colour a higher loss ratios for events with pre-season pre-events.

For events with hurricane force ($> 700\mathrm{N/m}^2$) the loss ratio and thereby the vulnerability is always reduced if the pre-event occurred at least one summer season in between and not within the same winter season irrespective of the intensity of the
pre-event.

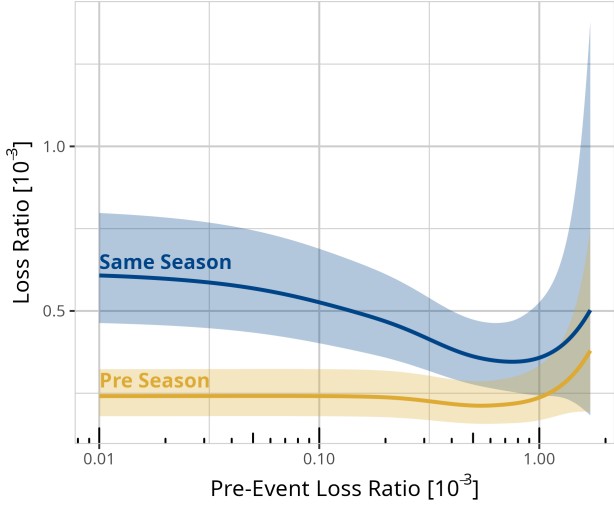

**Figure 6.** Comparison of events with a pre-event happening within the same season (blue) or with at least one summer season in between (yellow) for a fixed event of $750\mathrm{N/m}^2$. Shaded area indicates the 95% confidence interval.

Finally, we evaluate separately the impact of pre-events on the vulnerability for events with (blue) a pre-event occurring within the same season as the event and (yellow) the pre-event occurring at least one winter season before the event. Figure 6 shows the results for an event with a wind load of $750\mathrm{N/m^2}$ (Hurricane - 12 Beaufort). For pre-season pre-events, there is no significant effect with increase of pre-events loss ratios. On the other hand, events with same-season pre-events show a decrease in loss ratios with increasing pre-events' loss ratios up to around 0.8‰.

### 4.2.2 Weeks

While in the previous subsection the focus was on the impact of different winter half-years, here only events happening within the same winter half as the pre-event are taken into account. One week between event and pre-event is the week with the highest number of combinations (17% - Figure C1). The longer the time period between the pre-event and the event, the lower is the number of occurrences.

Figures 7a – 7c show the results of model $M_{Weeks}$ for three different event types, a gale (Figure 7c – $250\mathrm{N/m^2}$), a storm (Figure 7b – $500\mathrm{N/m^2}$) and a hurricane event (Figure 7c – $750\mathrm{N/m^2}$). For each event type, the different pre-event loss ratios are shown with a minor pre-event of 0.01‰ loss ratio, a medium pre-event of 0.1‰ and a major pre-event with 1‰.

Each of the nine combinations shows a decrease in vulnerability within the first two to three weeks as the loss ratio decreases with more weeks in between. In the case of storm (Figure 7b) and hurricane events (Figure 7c), a more intense pre-event leads to an increase in vulnerability if the two events occur within one week.

With increasing time between gale and storm events, the vulnerability is lower if the pre-event had a higher loss ratio. The high peaks of loss ratios in cases with eleven weeks between the previous and the current event could be explained by two events (Niklas – 31.03.2015 and Friederike – 18.01.2018), which made up nearly half of all data for this amount of weeks between events. These were two events with major damage, while the wind load was mostly in the range of a storm event at around 500 to $600\mathrm{N/m^2}$.

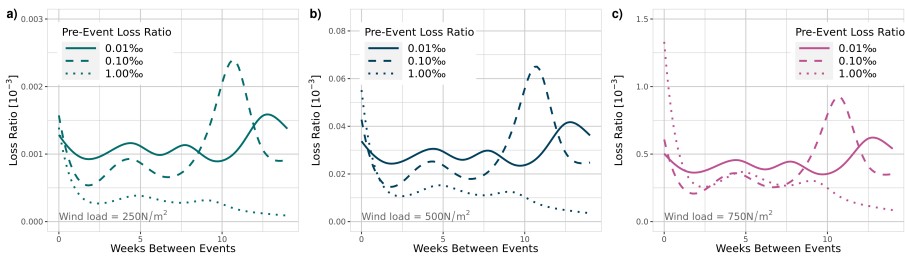

**Figure 7.** Results Model $M_{Weeks}$ for a fixed event of a) $250\mathrm{N/m^2}$, b) $500\mathrm{N/m^2}$ and c) $750\mathrm{N/m^2}$ and three different loss ratio intensities of pre-events (dotted lines 1‰, solid lines = 0.1‰, dashed lines = 0.01‰).

The differences between the pre-event loss ratios for each week between is not significant. However, the decrease in vulnerability is significant for events with pre-event loss ratios of 0.1‰ and one or more weeks in between on the 5% significant level

(two or more weeks on the 1% level) compared to events happening within the same week. (Figures D1 – D9 show different confidence intervals for each of the nine curves from Figures 7a – 7c)

### 4.2.3 Days

A total of 9.266 pre-events occurred within the 28 days preceding the event. Of these events, only 94 happened with only one full day between them. Most pre-events occurred 10 days before the event and two-thirds of all events occurred within the first two weeks (Figure C2).

The results of model $M_{Days}$ (Figures 8a – 8c) confirm the model $M_{Weeks}$ results of a steep decrease in vulnerability with increasing time between two events within the first weeks, and provide the opportunity to analyse the decrease at a higher temporal resolution.

For medium pre-events (pre-event loss ratio of 0.1‰) and major pre-events (pre-event loss ratio of 1‰) the decrease within the first days is stronger than for minor pre-events (pre-event loss ratio of 0.01‰). It takes eight to ten days until the vulnerability remains constant with increasing time between the events. These trends are significant for medium pre-events. For the case of a gale event, four days between the event and the pre-event leads to a statistically significant lower vulnerability (5% significance level) compared to events with only one day in between (Figure D2). For storm events, a significant difference can be found out to five days between the events (Figure D5). For hurricane events, the decrease is not statistically significant (Figure D8).

For the case of a gale (Figure 8a) or storm event (Figure 8b) the vulnerability of residential buildings increases with increasing pre-event loss ratio directly after the pre-event. Beyond about five days in between two events, a turning point emerges and the more intense the pre-event is, the less vulnerable is the affected district. The model results for a hurricane shows an unexpected result: the vulnerability of a pre-event loss ratio of 1‰ is between the vulnerability of a pre-event loss ratio of 0.01‰ and 0.1‰ for the first 3-4 days. Although it is not possible to fully resolve this issue, the assumption is that the lack of data for hurricane events with pre-event loss ratios of around 1‰ occurring just a few days before the event, leads to high uncertainties in the model estimation. Less than 0.1% (45 combinations in total) of the entire data set has a hurricane event and a pre-event with a loss ratio greater than 0.5‰.

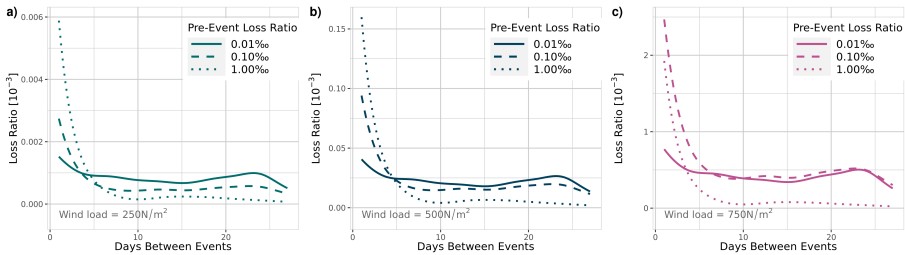

**Figure 8.** Results Model $M_{Days}$ for a fixed event of a) 250N/m$^2$ (gale), b) 500N/m$^2$ (storm) and c) 750N/m$^2$ (hurricane) and three different loss ratio intensities of pre-events (dotted lines = 1‰, solid lines = 0.1‰, dashed lines = 0.01‰).

## 5 Discussion

This study has assessed the temporal dynamics in the vulnerability of residential buildings to wind storm events in Germany based on insurance data. The focus is on the dependence of wind storm losses on the impact of previous storms, in terms of damage and timing. We see evidence for a decrease in vulnerability with increasing damage caused by the pre-event and increasing time elapsed since the pre-event. This is in line with studies of temporal dynamic vulnerability (Aerts et al., 2018; de Ruiter and van Loon, 2022). To our knowledge, for storm events an analysis of the influence of pre-events on vulnerability has not yet been conducted.

### 5.1 Characteristics of the insurance data

The daily temporal resolution within the 23-year dataset enables a detailed analysis of the time intervals between events, providing the opportunity to evaluate consecutive storm events occurring within days or weeks. This combination of high temporal resolution and an extensive dataset offers a unique opportunity to explore and understand the temporal dynamics of vulnerability. However, note that the report of damage to the insurance company does not always occur directly after the causing event, leading to potential misassignments of the reported damage to the wrong event. Late reported damage might get assigned to the more intense pre-event, especially for weak events following a very intense event within one season. This misassignment could increase the loss assigned to the previous intense event and decrease the loss assigned to the weak following event. A misassignment is unlikely when there is a whole summer season in between events. Therefore, minor events with pre-season pre-events might have a higher vulnerability than minor events with same-season pre-events.

As the losses given in the insurance data set are on the aggregation level of districts, effects on the vulnerability can only be quantified on the very same level of spatial aggregation. Especially for weak pre-events (weak in terms of wind load and hence losses), the probability is lower that the same building is hit by both events. A finer spatial resolution would help reduce this effect. Another illuminating approach would be to derive different vulnerability curves for various building types, provided this information were to be available (Smith and Henderson, 2016).

Excluding summer half-year from our analysis is a necessary choice, but leads to further uncertainties. For events with associated pre-events occurring not in the same season but with one (or more) summer season in between, a summer hail event or other hazard might damage buildings; this leads to a reduced vulnerability. In our approach, this decrease of vulnerability is attributed to the previous storm event. For future work, it would be desirable to have a damage data set in which hail- and storm-related damage can be distinguished, and to extend our model approach from single- to multi-hazard events in order to analyze the temporal dynamics in a more holistic way.

### 5.2 Extensions and transferability

Including the three covariates – *pre-event loss ratio*, *time between events*, and $mV_{1914}$ – in our model is appropriate to analyze the temporal dynamics of vulnerability. However, vulnerability is a complex construct and additional factors have not yet been included. Two factors influencing physical vulnerability of residential buildings to be investigated in the future are "extreme

event warnings" and the spatial variability of vulnerability. The former is necessary to investigate if preparatory action (e.g. closing windows, clearing plant pots from the balcony or retracting awnings) related to warnings reduce the vulnerability to an event. A comprehensive analysis of spatial variability is required to make more reliable statements about the transferability of the results. For Germany, four different wind load zones with different building standards have been defined, depending on the 10-min average wind speed (ENV-1991-24, 1995; DIN 1055, 2005). In addition, some federal states have their own specific insurance policies, such as deductibles (Baden-Württemberg). Without knowing the effect of these standards, a transfer of the results is not recommended. However, it is desirable to use our approach for similar data sets from other countries and/or other hazards to compare the results and learn from the similarities and differences. For future work, the non-linear relations between the *time between two events* and the *loss ratio* justifies the advantage of generalized additive models over generalized linear models. Additionally, our model approach provides an excellent starting point for quantifying the temporal dynamics of vulnerability. It offers a flexible framework, where incorporating additional predictors or replacing existing ones can be accomplished with minimal effort. This adaptability makes it a valuable tool for further refinement and application to other contexts.

In the case of risk assessments related to climate change and hence long-term trends, ignoring the results of decreasing vulnerability and assuming stationary vulnerability over time would lead to an overestimation of the risk. We thereby confirm the theories by Aerts et al. (2018) and de Ruiter and van Loon (2022). A direct transfer of our results to risk assessments for the future should be considered with caution, as other factors such as changes in building standards or construction techniques can also reduce vulnerability.

### 5.3 Underlying processes in temporal dynamics

For the daily and weekly scale we find: first, vulnerability decreases with increasing time between an event and its pre-event; the decrease is strongest within the first 10 days. This indicates that in Germany, a huge amount of reconstruction is done in that short time. Second, for a time between events of $\lesssim 5$ days, the vulnerability increases with increasing loss of the pre-event (Figures E1 – E3). One possible explanation for the observed increase in vulnerability is that buildings already damaged by the initial event are more susceptible to further damage in subsequent events. For instance, roof tiles that have been loosened or windows that have been broken may not have been repaired within the short time span before the following event, increasing the likelihood of additional damage. While it could be argued that elements such as broken windows or fallen roof tiles cannot be damaged again and may therefore reduce the overall vulnerability, we assume that prior external damage increases the risk of subsequent internal damage. For example, a missing roof tile can significantly raise the probability of water intrusion during the next storm, potentially leading to more severe interior damage. In contrast, certain damage mechanisms, such as fallen trees impacting buildings, represent one-time events that do not contribute to increased vulnerability in subsequent hazards. However, the influence of such individual cases cannot be quantified within the scope of this study, as the available data only provides district-level spatial resolution. On district level the increase in vulnerability with increasing pre-event loss ratio on short term can be attributed either to a growing number of damaged buildings resulting from the initial event, to a higher loss

ratio within individual buildings due to pre-existing damage, or to a combination of both factors—with the latter being the most plausible explanation.

## 6  Summary and Conclusions

We refer to the concept of temporal vulnerability which has been discussed in the literature (e.g. Papathoma-Köhle et al., 2012; Di Baldassarre et al., 2018). To our knowledge, this study is the first to suggest a quantitative model for the physical vulnerability to wind storm events in Germany and it's dependence on the characteristics of a previous event, i.e. quantifying temporal dynamic vulnerability. We focus active on intensity of previous events and the time between the two events. With generalized additive models, we describe the non-linear a potentially non-linear functional relationships between the loss ratio and the time between events, as well as the intensity of the pre-event.

The key findings of this study indicate that vulnerability decreases with increasing time intervals between two events across daily, weekly, and seasonal timescales, with the most substantial reduction observed within the first 10 days. Moreover, vulnerability increases with the severity of the preceding event, although this effect is confined to short inter-event periods of less than approximately five days. Notably, for time intervals greater than five days, higher loss ratios from preceding events are associated with decreased vulnerability in subsequent events. Both findings are in line with the theories of Aerts et al. (2018); de Ruiter and van Loon (2022).

This model helps to quantify vulnerability in risk assessment, possibly leading to an improved understanding of past events' damages and the resulting losses. The resulting more accurate predictions of loss or risk are instrumental in enhancing future risk management and could therefore prove beneficial to insurance and reinsurance companies. Our findings underscore that vulnerability is influenced by the timing and intensity of previous events, highlighting the necessity of treating vulnerability as a temporally dynamic parameter in risk modelling frameworks.

*Code availability.*

*Data availability.* Due to the data protection policies of the data provider German Insurance Association, the data cannot be made available.

*Code and data availability.*

*Sample availability.*

*Video supplement.*

## Appendix A

400 This part of the appendix includes additional information on the method of spatial allocation of meteorological and insurance data (subsection 3.3). The three figures are intended to make it easier to understand the allocation using examples. Figure A1 and Figure A2 explain the choice of using a buffer around districts for the allocation in general, while Figure A3 gives an example for the choice for the buffer of 31km.

### A1

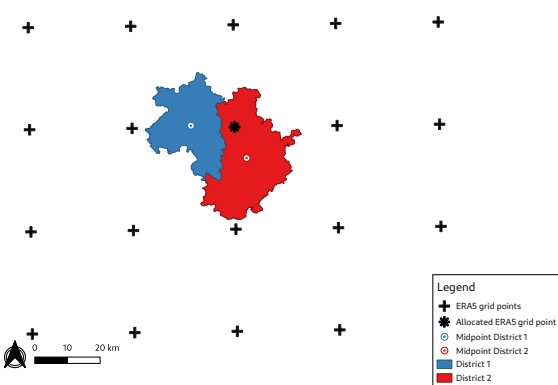

**Figure A1.** Example for allocation of meteorological data (ERA5) and damage data (insurance data on district level) as conducted by Pardowitz et al. (2016) and Donat et al. (2011a). First the district midpoint is calculated and then assigned to the nearest reanalysis grid point. In some cases (as shown in this example) it leads that two district are assigned to the same grid point for each time step.

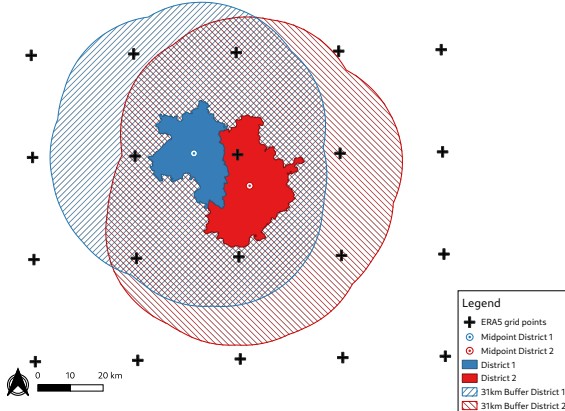

**Figure A2.** Example for allocation of meteorological data (ERA5) and damage data (insurance data on district level) as conducted in this study. First a buffer of 31km is calculated for each district. Then all grid points in this buffer are assigned to the district. Last, for each time step the maximum of all assigned grid points is taken for the next step in the modelling approach. Thereby two districts do not have the same grid point allocation in every time step.

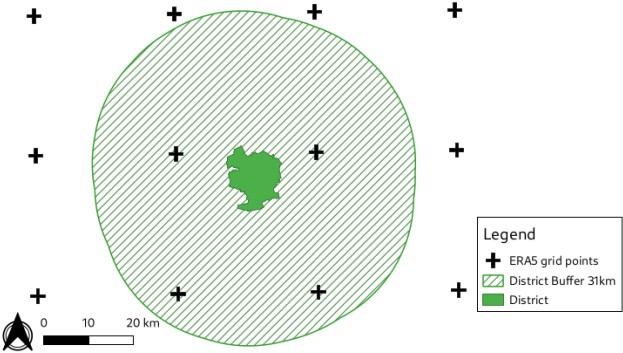

**Figure A3.** Example to illustrate the choice of 31km for the buffer zone. By choosing a buffer of 31km we can assure that for each district at least four grid point are taken into account for the allocation.

## Appendix B

This part of the appendix includes detailed information and results of the covariate $mV_{1914}$.

### B1

The additional covariate *mean value 1914* ($mV_{1914}$) is included in the model approach to account for the overall development of buildings within a district. These overall developments are described in the introduction as changes in vulnerability due to non-hazard factors. Trends, such as higher building standards for new buildings, could lead to a decrease in overall vulnerability in a district, without an event occurring. If these trends are not captured with an additional covariate in the models, the implemented covariates take these trends into account and would not represent their purpose, leading to possible misinterpretations. Therefore, we use with the mean value 1914 ($mV_{1914}$)

$$mV_{1914} = \frac{\text{Total Value 1914 per district in €}}{\text{Number of Insurance Contracts per district}}, \tag{B1}$$

a covariate that represents the non-hazard specific changes of the vulnerability and is used as a baseline. The value 1914 ("Wert 1914") is a fictive value for insurance companies in Germany, reflecting the value a building would have cost in gold mark in the year 1914 (GDV, 2024). For better comparability, residential buildings are valued in values 1914 by insurances, as in this year, construction costs were not subject to any significant fluctuations. Dividing the 1914 value by the number of insurance contracts in a district, we obtain an approximation for the average standard of residential buildings in a district. It is assumed that an increase in $mV_{1914}$, leads to a decrease in vulnerability. It should be noted that the value 1914 includes not only the building quality, but also the building type (e.g. detached house or apartment building).

### B2

The results show a decrease in vulnerability with increasing *mean value 1914*. These findings are in line with the assumption that better building conditions are related to a lower vulnerability. However, taking *mean value 1914* as a proxy for non-hazard-specific changes in vulnerability is a simple approach, and a detailed analysis of factors influencing these changes, as well as a higher spatial resolution for these changes, is desirable.

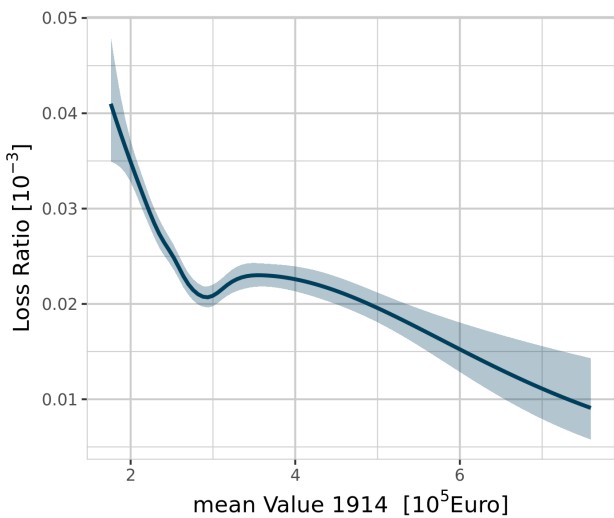

**Figure B1.** Results of $M_{preEvent}$ for the covariate $mV_{1914}$ for an event with wind load = $500\mathrm{N/m}^2$ and a pre-event loss ratio of 0.05‰

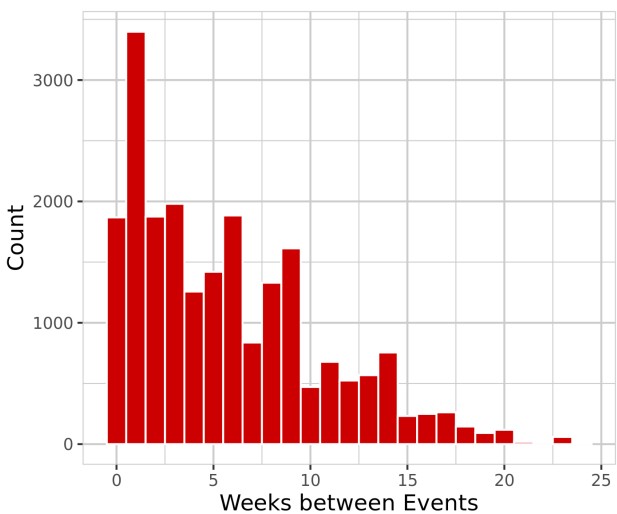

**Figure C1.** Histogram for pre-events occurring within the same season as the event accumulated to weekly basis.

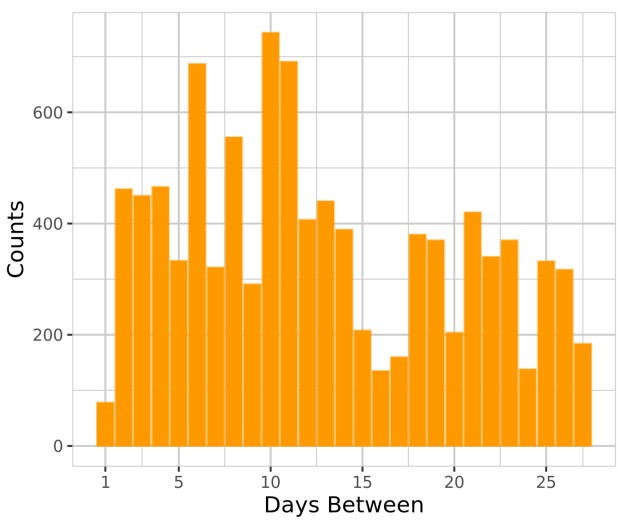

**Figure C2.** Histogram for pre-events occurring within 28 days before the event.

**Appendix D: Significance analysis**

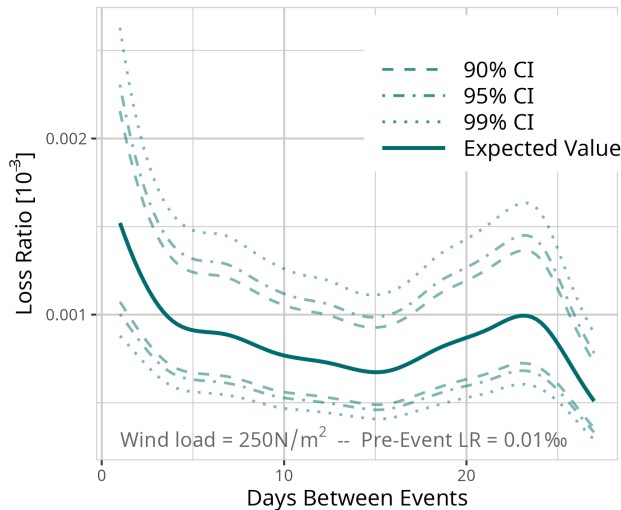

**Figure D1.** Expected Value based on the results of Model $M_{Days}$ for a fixed event of $250\text{N}/\text{m}^2$ and a pre-event loss ratio of 0.01‰. Confidence intervals (CI) (90% = dashed, 95% = dot-dashed and 99% dotted line) are used for significance analysis.

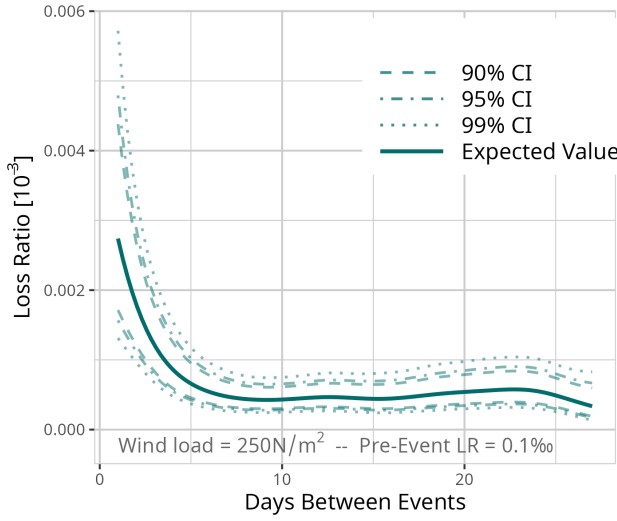

**Figure D2.** Expected Value based on the results of Model $M_{Days}$ for a fixed event of $250\text{N}/\text{m}^2$ and a pre-event loss ratio of 0.1‰. Confidence intervals (CI) (90% = dashed, 95% = dot-dashed and 99% dotted line) are used for significance analysis.

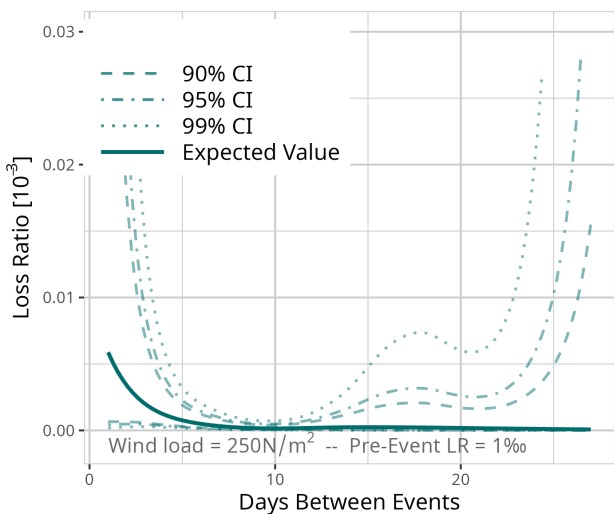

**Figure D3.** Expected Value based on the results of Model $M_{Days}$ for a fixed event of $250\text{N}/\text{m}^2$ and a pre-event loss ratio of 1‰. Confidence intervals (CI) (90% = dashed, 95% = dot-dashed and 99% dotted line) are used for significance analysis.

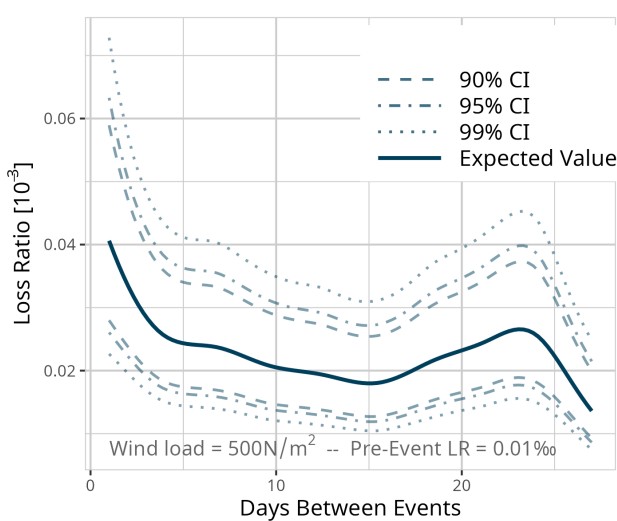

**Figure D4.** Expected Value based on the results of Model $M_{Days}$ for a fixed event of $500\text{N}/\text{m}^2$ and a pre-event loss ratio of 0.01‰. Confidence intervals (CI) (90% = dashed, 95% = dot-dashed and 99% dotted line) are used for significance analysis.

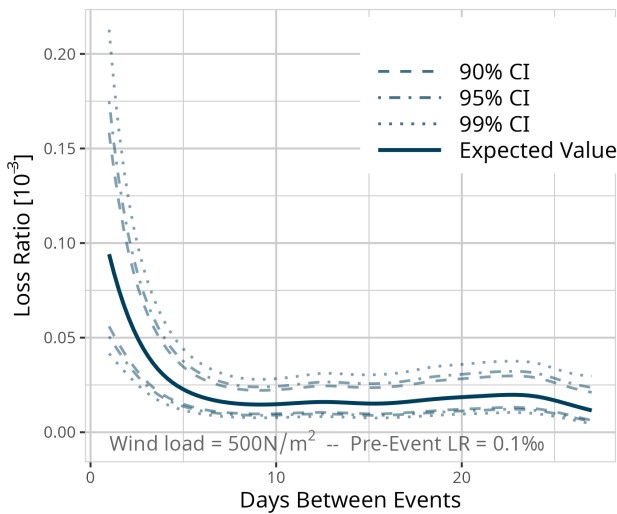

**Figure D5.** Expected Value based on the results of Model $M_{Days}$ for a fixed event of $500N/m^2$ and a pre-event loss ratio of 0.1‰. Confidence intervals (CI) (90% = dashed, 95% = dot-dashed and 99% dotted line) are used for significance analysis.

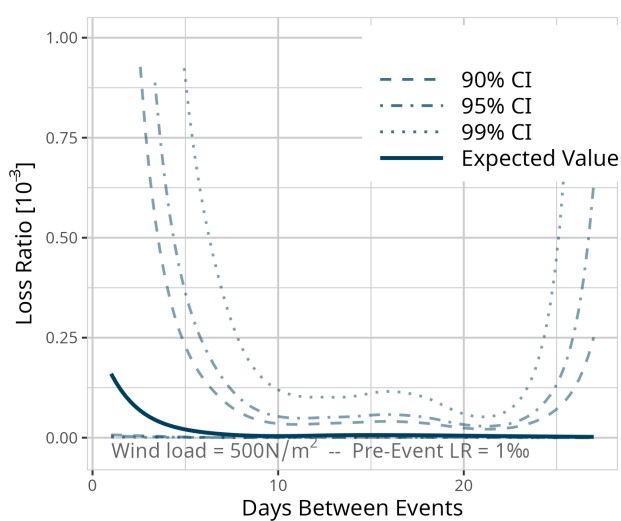

**Figure D6.** Expected Value based on the results of Model $M_{Days}$ for a fixed event of $500N/m^2$ and a pre-event loss ratio of 1‰. Confidence intervals (CI) (90% = dashed, 95% = dot-dashed and 99% dotted line) are used for significance analysis.

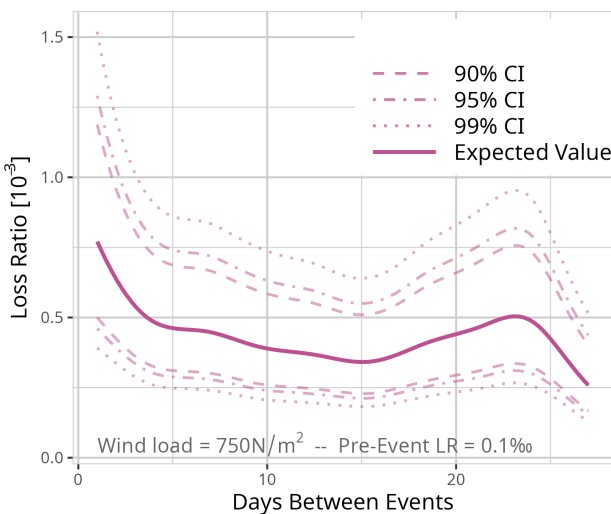

**Figure D7.** Expected Value based on the results of Model $M_{Days}$ for a fixed event of $750\text{N}/\text{m}^2$ and a pre-event loss ratio of $0.01‰$. Confidence intervals (CI) (90% = dashed, 95% = dot-dashed and 99% dotted line) are used for significance analysis.

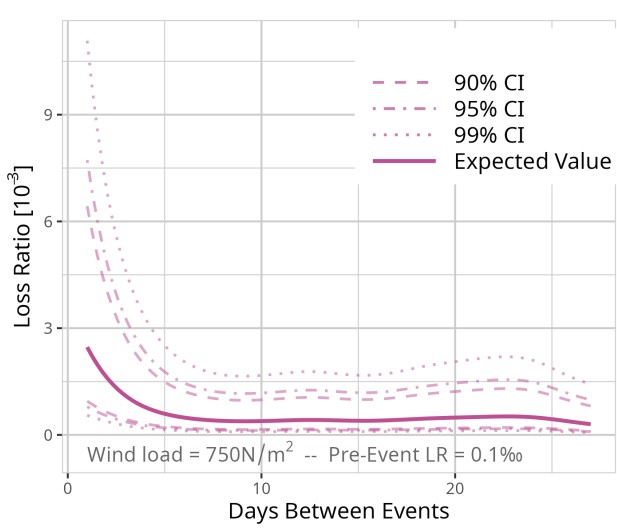

**Figure D8.** Expected Value based on the results of Model $M_{Days}$ for a fixed event of $750\text{N}/\text{m}^2$ and a pre-event loss ratio of $0.1‰$. Confidence intervals (CI) (90% = dashed, 95% = dot-dashed and 99% dotted line) are used for significance analysis.

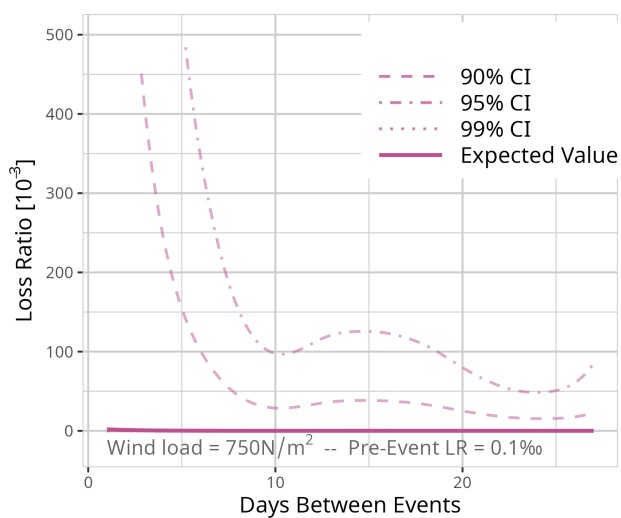

**Figure D9.** Expected Value based on the results of Model $M_{Days}$ for a fixed event of $750\text{N}/\text{m}^2$ and a pre-event loss ratio of 1‰. Confidence intervals (CI) (90% = dashed, 95% = dot-dashed and 99% dotted line) are used for significance analysis.

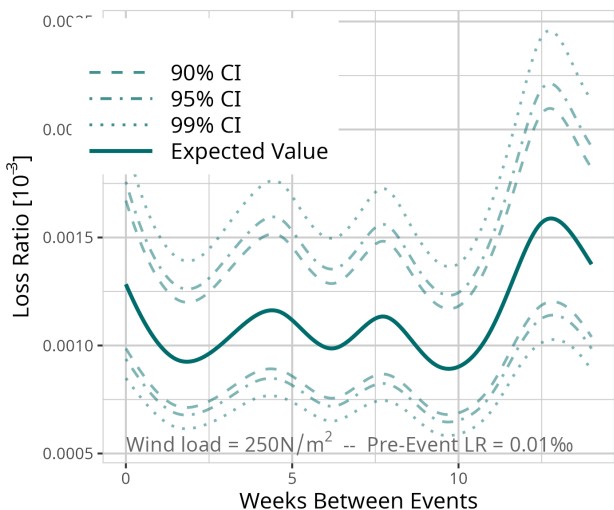

**Figure D10.** Expected Value based on the results of Model $M_{Weeks}$ for a fixed event of $250\mathrm{N/m^2}$ and a pre-event loss ratio of 0.01‰. Confidence intervals (CI) (90% = dashed, 95% = dot-dashed and 99% dotted line) are used for significance analysis.

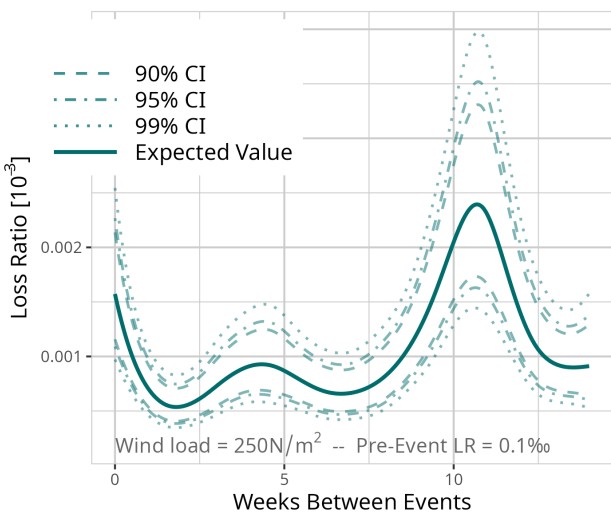

**Figure D11.** Expected Value based on the results of Model $M_{Weeks}$ for a fixed event of $250\mathrm{N/m^2}$ and a pre-event loss ratio of 0.1‰. Confidence intervals (CI) (90% = dashed, 95% = dot-dashed and 99% dotted line) are used for significance analysis.

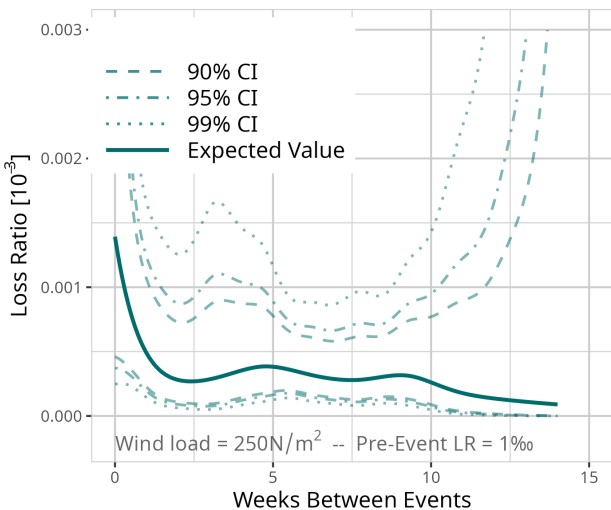

**Figure D12.** Expected Value based on the results of Model $M_{Weeks}$ for a fixed event of $250\mathrm{N/m}^2$ and a pre-event loss ratio of 1‰. Confidence intervals (CI) (90% = dashed, 95% = dot-dashed and 99% dotted line) are used for significance analysis.

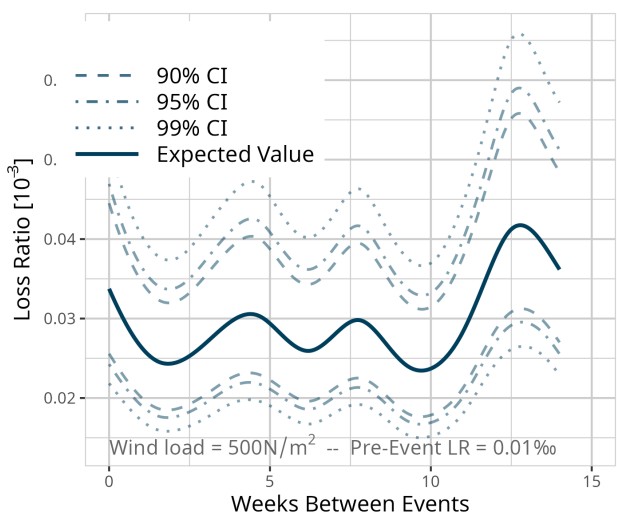

**Figure D13.** Expected Value based on the results of Model $M_{Weeks}$ for a fixed event of $500\mathrm{N/m}^2$ and a pre-event loss ratio of 0.01‰. Confidence intervals (CI) (90% = dashed, 95% = dot-dashed and 99% dotted line) are used for significance analysis.

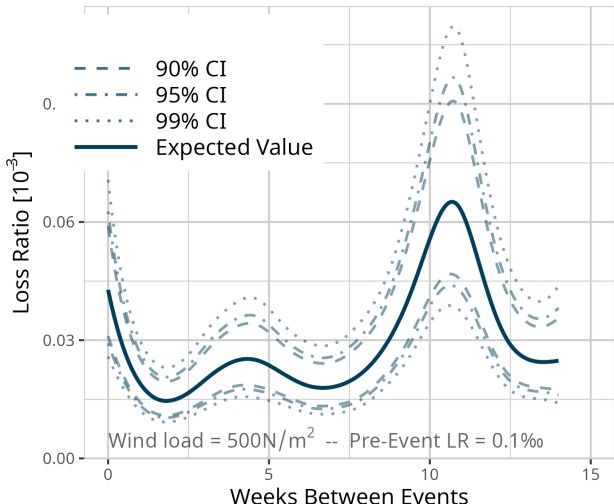

**Figure D14.** Expected Value based on the results of Model $M_{Weeks}$ for a fixed event of $500\mathrm{N/m}^2$ and a pre-event loss ratio of 0.1‰. Confidence intervals (CI) (90% = dashed, 95% = dot-dashed and 99% dotted line) are used for significance analysis.

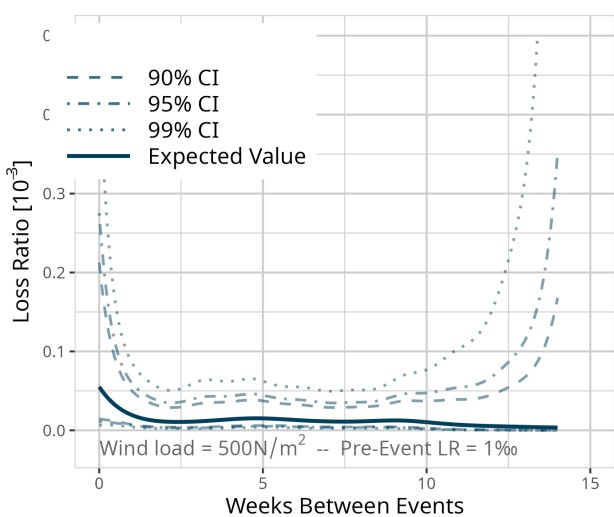

**Figure D15.** Expected Value based on the results of Model $M_{Weeks}$ for a fixed event of $500\mathrm{N/m}^2$ and a pre-event loss ratio of 1‰. Confidence intervals (CI) (90% = dashed, 95% = dot-dashed and 99% dotted line) are used for significance analysis.

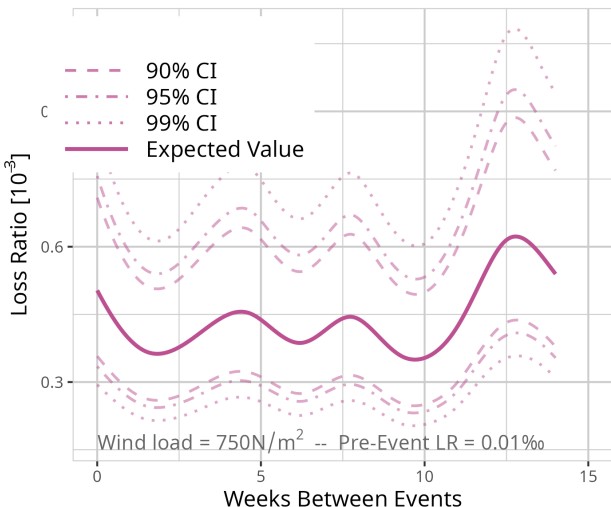

**Figure D16.** Expected Value based on the results of Model $M_{Weeks}$ for a fixed event of $750\text{N}/\text{m}^2$ and a pre-event loss ratio of 0.01‰. Confidence intervals (CI) (90% = dashed, 95% = dot-dashed and 99% dotted line) are used for significance analysis.

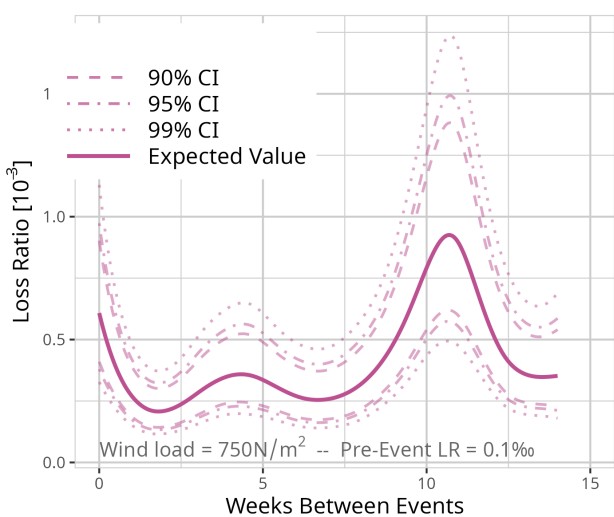

**Figure D17.** Expected Value based on the results of Model $M_{Weeks}$ for a fixed event of $750\text{N}/\text{m}^2$ and a pre-event loss ratio of 0.1‰. Confidence intervals (CI) (90% = dashed, 95% = dot-dashed and 99% dotted line) are used for significance analysis.

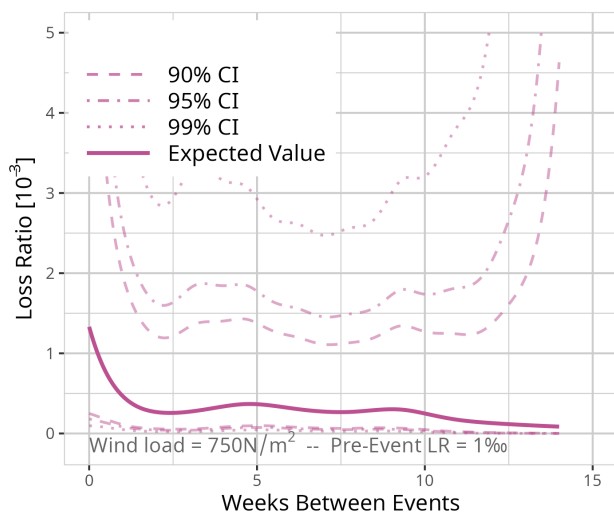

**Figure D18.** Expected Value based on the results of Model $M_{Weeks}$ for a fixed event of $750\text{N}/\text{m}^2$ and a pre-event loss ratio of $1‰$. Confidence intervals (CI) (90% = dashed, 95% = dot-dashed and 99% dotted line) are used for significance analysis.

## Appendix E

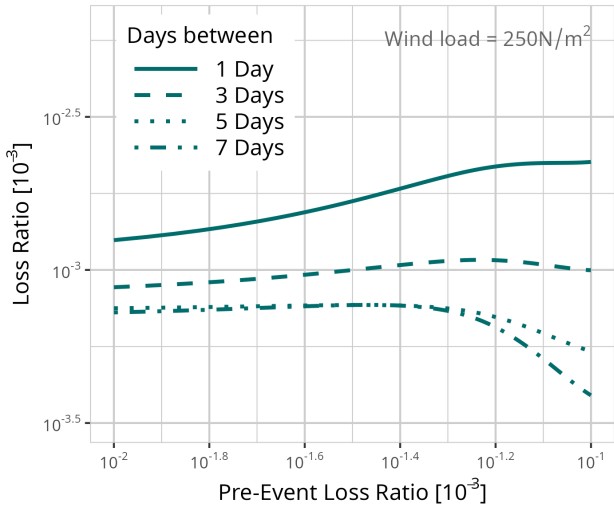

**Figure E1.** Expected Value based on the results of Model $M_{Days}$ for a fixed event of $250\mathrm{N/m^2}$. Line-types depict different time periods between the events.

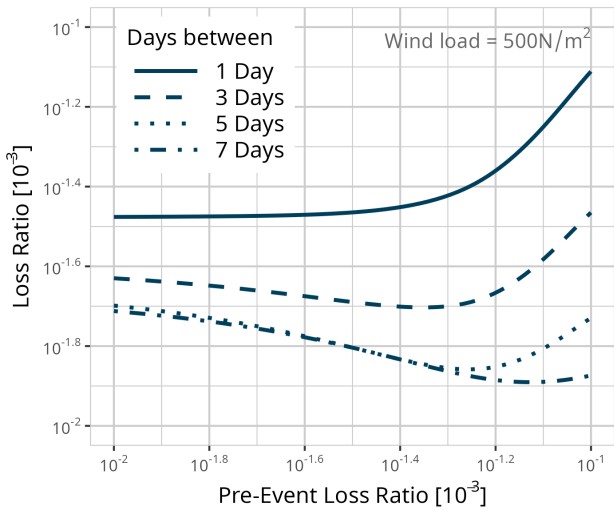

**Figure E2.** Expected Value based on the results of Model $M_{Days}$ for a fixed event of $500\mathrm{N/m^2}$. Line-types depict different time periods between the events.

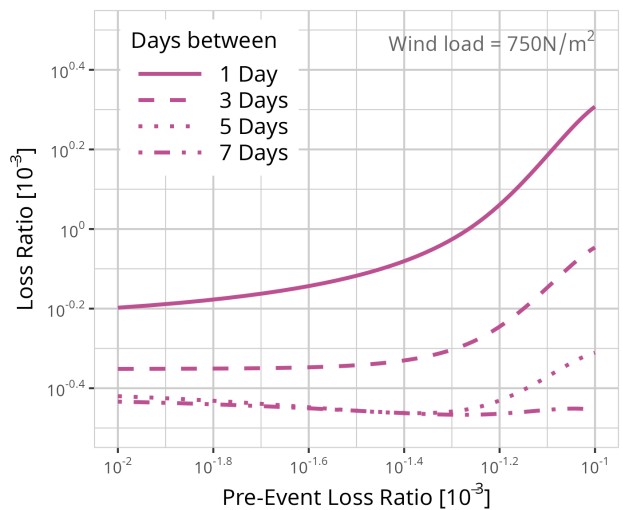

**Figure E3.** Expected Value based on the results of Model $M_{Days}$ for a fixed event of $250\text{N/m}^2$. Line-types depict different time periods between the events.

*Author contributions.*  Andreas Trojand: Conceptualization, Data curation, Formal analysis, Methodology, Software, Visualization, Writing – original draft preparation, review & editing

Henning Rust: Conceptualization, Methodology, Supervision, Funding acquisition, Writing – review & editing

Uwe Ulbrich: Conceptualization, Supervision, Funding acquisition, Writing – review & editing

*Competing interests.*  One of the co-author is a member of the editorial board of Natural Hazards and Earth System Sciences.

*Acknowledgements.*  The research presented in this article was conducted within the research training group "Natural Hazards and Risks in a Changing World" (NatRiskChange) funded by the Deutsche Forschungsgemeinschaft (DFG; GRK 2043/2). We thank the *Gesamtverband der Deutschen Versicherungswirtschaft e.V.* (GDV) for providing the loss data and we are grateful to Tristian Stolte and one anonymous referees for providing valuable comments to improve the manuscript and to Edmund Meredith for proofreading the manuscript.

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
