# Peer review of "Temporal dynamic vulnerability - Impact of antecedent events on residential building losses to wind storm events in Germany"

_EGUsphere, 2024_

## Author Response (AR1)

The document is an extension of the author's response of February 5 and February 6, 2025 to the reviewers' comments. After we have adapted the manuscript, we can address some of the reviewers' comments in more detail.

**Response to Referee #1 Tristian Stolte**

Thank you for carefully reading the manuscript and for your very positive assessment of our work's potential! We very much appreciate your detailed comments and helpful questions.

Regarding the specific comments:

*Introduction*

- *l21 & 23: Natural disasters or natural catastrophes do not exist. A disaster only is a disaster when humans or human objects are involved. Alternatively, the authors could use terms like "natural hazards", "disasters from natural hazards", or simply "disasters" (https://www.undrr.org/our-impact/campaigns/no-natural-disasters).*

  - This is a good point! We changed the concepts/wording accordingly.

- *I am glad to see that vulnerability is explicitly defined by the authors (l40-41). I did expect it earlier, where the authors first refer to the risk framework of the IPCC and UNDRR (l27-28). This may be a personal preference though, but I would like to ask the authors to reconsider the timing of presenting the reader with the definition of their main topic.*
  - Thank you, we moved the definition to line 27-29. Directly after the definition of risk.

- *The two sentences below sound counterintuitive. Do the authors mean that buildings experience moderate damage, but that the accumulated damage is nonetheless large? Please specify. 1. "Although individual natural disasters, such as the flash flood that occurred in western Germany in July 2021 or the Elbe flood in 2002, are the most devastating single events, the accumulated damage from storm events caused three times more damage to residential buildings than other natural disasters between the years 2002 and 2021 (GDV, 2023)." (l20-24) 2. "The impact on human life is relatively small, and the damage to individual buildings is generally moderate." (l24-25)*
  - Thanks for pointing this out. We can understand that this part sounds counter-intuitive and have therefore reworded it.

- *L49-53: the authors mention several building-related drivers of vulnerability, which they group under 'non -hazard specific dynamics; . Then they refer to Drakes & Tate (2022) and Simpson et. al. (2021), which are both on social vulnerability. And which do not have a clear link to buildings or the way in which non-hazard specific dynamics are portrayed here. Maybe the authors can try to solidify this bridge.*
  - Yes, this is a valid point. we revised this part.

- *L71-72: "The time between two events is crucial for winter storm events in Germany as these occur in a short amount of time more often than other hazards." -> This sentence needs a*

*reference. If Mühr et al. (2022) compare the frequency of winter storms with other hazards, then please make it explicit.*

- Thanks! A comparison with other hazards was not meaningful at this point. We just want to show that storms can occur within a short period of time. Mühr et. al. is a source for this

*Data*

- *Depending on the journal's requirements, the authors may be able to merge the Data and Methods sections, as there is no clear distinction between the two in terms of topic. I would say that the data processing is also part of the Methods, and that several data-related choices (like the pre-event threshold) are made in the Methods. It may also make more sense to combine sections 2.2 and 3.1 as I only understood why the authors added Table 1 when I arrived at section 3.1.*
  - Thanks for pointing us towards this irritation. We have adapted sections 2.2. and 3.1. so that the irritation will hopefully no longer occur.

- *L92-93: "…the damage is almost exclusively caused by windstorm events." -> could the authors provide a source for this? Or is this in the insurance data?*
  - The insurance data covers losses related to contracts covering hail and storm. It is, however, not possible to derive the cause from the insurance data. A popular approximation for separating losses due to hail and due to storm is to separate the data set into a summer and a winter data set as hail is related to thunderstorms, a hazard predominantly occurring in summer and rarely in winter; winter storms are a winter phenomenon. We adapted this in the text.

- *In the loss ratio (eq. 1), why would you multiply the insured sum by $10^3$? If this is just the label from the insurance data, then I would suggest to leave the $10^3$ part out as it is a bit confusing form a mathematical standpoint.*
  - Yes, indeed, this is not a clear way of presenting the equation. The loss ratio is, however, typically given in euros loss per thousand euros insured sum. We changed the equation to a mathematical sound one.
  -
- *Could the authors include the formula for air density?*
  - The denstiy of air is either approximated via the ideal gas law of dry air or as the density of a mixture of two ideal gases: dry air and water vapour. Both partial densities can be obtained from the ideal gas law. We added this in the manuscript

- *Could the authors explain why it is important to include the "orography due to air density" (l108) into account?*
  - Thanks for pointing this out! We changed the sentence and made it clearer why it is import in the lines 124 to 127.
  -

- *I do not follow the argumentation on why the authors buffer the districts with 31km buffers such that four ERA5 grid points can fit in, because the authors mention that the ERA5 also has a 31km resolution. A flowchart with some visuals can clearly demarcate such steps and help the reader to follow along with spatial data analysis. This also helps understanding the rest of the methods.*
  - The procedure is a way to include grid points with centers just outside the districts but with their area reaching into the districts. We explain it in the revised manuscript in more detail and give three figures in the appendix for a better understanding.

*Methods*

- *In Figure 1 -> is an event one windstorm and the total loss ratio per district? Or is this defined differently? Please make this more explicit.*
  - For the event definition, we follow the threshold approach of the German homeowner insurance. For an event, this threshold has to be exceeded by the maximum wind speed over all grid points assigned to the district. The loss ratios is given per district. We made this more explicit in the caption of figure 1.

- *If pre-event (as mentioned in the text) = previous event (as mentioned in Figure 1), then please make this explicit.*
  - Thanks! Point taken and we changed it to a consistent naming of concepts in the revised version.
  -
- *I am not expert in GAMs, so it would be good to let someone else review this.*
  - We appreciate your honesty! A second reviewer commented on that section.

- *L146-147: "Recent studies include only the hazard component and a parameter for exposure as covariates into the damage model approach (e.g. Heneka et al., 2006; Pardowitz et al., 2016; Welker et al., 2021)." Heneka et al. (2006) is not really a recent study anymore and Pardowitz et al. (2016) is also a bit on the edge of being recent. Maybe this can be rephrased to 'past studies'?*

  - We changed accordingly. Thanks!

- *L157-158: "In model MDays the temporal scale is refined further to a diurnal time scale and only pre-events happened within four weeks (28 days) before the event are included in the model." -> The authors seem to confuse the terms special scale with spatial resolution and use them interchangeably. The temporal scale is 28days, but the temporal resolution is diurnal, is that correct? See also: (l159-160) "These different temporal resolutions are chosen because shortly after the event occurrence, we expect the daily scale being essential…".*
  - We are not sure if we understand your question. The lines 157-158 do not mention any spatial scales. Also, we do not change the temporal resolution of the data (e.g. by aggregating); we do, however, create covariates which address previous events on time

scales of either days (M_Days) or weeks (M_Week) or seasons (M_season). We realize, however, that this is not easy to understand and we have rewritten this paragraph.

- *L159-160: "we expect the daily scale being essential and around one seventh of all previous events fall in this time span" -> why do the authors expect this and why one-seventh?*
    - Thanks, we explain the daily scale in more detail. The one-seventh, we left out in the revised version as it was not important from the beginning.

- *The part on mean value 1914 was a bit hard to follow, although I understand the covariate's purpose and the reasoning. What I don't understand yet is what is the "number of contracts". Contracts for what? Insurance contracts? And what does that mean for the m1914 value? Can the authors please make this more explicit.*
    - This is indeed a very difficult part and we agree that the description needs another iteration. Yes, we are talking about insurance contracts and as values measured in € change e.g. due to inflation, the value 1914 is a reference used to compare values (and losses) across decades. We decided to keep the explanation within the text short, but have now a detailed part in the appendix about the value 1914

*Results*

- *L176-177: "Especially events with wind loads larger 700N/m$^2$ (Beaufort scale 12) and previous event loss ratios of more than 0.1‰ occur rarely." Why do the authors highlight these and what is meant by 'rarely'?*
    - We want to point out that there is a range of wind loads and losses which is very interesting for insurances but where data is sparse. The sentence needs more concrete information, like what is the ration of events over these thresholds. We have made it more explicit in the revised version.

- *L183-184: "Thus the vulnerability describing the loss ratio for a certain wind load is lower in cases with higher previous event loss ratios." -> do the authors mean The loss ratio, which describes the vulnerability, for a certain wind load is lower in cases with higher previous event loss ratios?*
    - Thank you for pointing this out. Our main focus here is on vulnerability and not on the loss ratio. However, the wording is ambiguous and we adjusted it in the revision.

- *L196: "…lead to a significantly lower vulnerability for events with a pre-season event than with a same-season event…" -> where do the authors base this significance on? Is it statistically significant? Or do they mean something like 'substantial'?*
    - In this study, significantly different always means based on statistical significance. The shaded areas depict 95% confidence for the estimated of the expected loss ratios. If confidence intervals for "same-season pre events" and "pre-season pre-events" do not overlap, we can safely consider the difference as statistically significant. In the revised version, we added text to clarify this.

- *Figures 7 and 9 could go to the appendix, maybe together with similar figures on the input data. The manuscript already has several interesting figures.*
  - Thanks for the advice and the compliment for the other figures. We took this into account in the review.

- *Figure 10c shows an oddity that is not discussed: why is the 1.00 per mille line in between the 0.1 and 0.01 per mille lines for the first 4-5 days? The final paragraph of section 4.2.3 discusses Figures 10a and 10b but not 10c.*
  - There are very few events where two such extreme events occur within 1-2 days of each other which leads to large uncertainties. We added the missing description in the revised version.

*Discussion*

- *L255-257: "Kreibich et al. (2023) focused on socio-hydrological data and used the Likert scale to estimate changes in the vulnerability, but, for example, did not quantify the impact of the time between two events." This example comes a bit out of the blue and seems not very relevant for the argumentation.*
  - We agree that this example is not suitable here and adapted it accordingly. Thanks

- *L270-274: "We consider only winter months for wind storm events and exclude summer months, as the damage data set includes storm and hail damages for residential buildings, but it is not possible to distinguish within the data between the causes for a certain loss. Similar to storm events, hail events mainly lead to roof and window damage of residential buildings. A hail event is likely to impact the vulnerability of residential buildings, which could not be included in this work due to the lack of hail data in the ERA5 data set and others. In general, extending the models to a multi hazard approach is desirable." -> This whole part sits better in the methods as it presents reasons of why the authors investigate winter storms and not summer storms.*
  - We moved and rewrote this part to the earlier section on methods for the revised version.

- *L275-277: "On a seasonal scale, the differences between same-season and pre-season pre-events increase strongly with exceeding a wind load of around 600N/m2 and thereby considerable structural damage according to the Beaufort scale. While on lower wind loads, only slight structural damages occur." -> this is a repetition of the results.*
  - Thanks for the point, we removed this paragraph from the discussion.

- *Overall, the discussion needs to be restructured. It is currently a collection of small paragraphs where the authors repeat results or briefly validate some methodological choices. Instead, it would be much more interesting to read about the implications of the results (which are very interesting!), like in the paragraph on l278 – 283. Also, more concrete advice on methods for follow-up studies could be presented. They could also talk about other factors that are important in (dynamic) vulnerability assessments, which are not included in this study. Another*

*suggestion would be to talk more about the transferability of these results. And what about spatial differences between the regions under study?*

- Thank you for these points! We restructured the discussion added your points and extended some other.

*Technical corrections*

*Overall, sentence structure is sometimes off. I would advice to let a native English speaker review the sentence structure to ensure that no unnecessary mistakes are made. Below are some examples, but not all of them are listed here.*

- Thank you very much for pointing this out and giving some example. We have not marked every technical improvement in the revised manuscript, but we have hopefully corrected all of them ourselves and thanks to a native English speaker.

**Response to Referee #2  - Anonymous**

Thank you for carefully reading the manuscript and for your positive assessment of the work's value! We very much appreciate your valuable suggestions. The revision edited language with the help of a native speaker to eliminate the language issues you pointed out.

Regarding your suggestions:

1. The claims in the Abstract should be supported by data.

- As far as points in the abstract are not self-evident, we added references in the abstract.

2. Lines 29 –34, The content of this paragraph is not closely related to the topic of this article. It is recommended to adjust it to enhance its relevance to the title.

- In the mentioned paragraph in the introduction, we tried to put our research into a wider frame of storm impact research and pointed towards popular research activities (storm damage and changes in storm damage) to show that our research topic has not been covered extensively. We find that this is adequate and hope we made this more explicit in the revised version.

3. The authors are advised to provide a summarizing analysis of the research gap at the end of the introduction and briefly introduce the research content.

- Thank you for this point. We added a summarizing analysis in the introduction.

4. Lines 91 –92, If the influence of storms from other seasons is excluded, the authors need to explain whether the impacts of storms from other seasons are similar to those of winter storms.

- Unfortunately, we have not well formulated that case. The other seasons are not excluded because they are similar but due to the ambiguity of the damage data in other seasons. The data does not contain information about the cause of the damage, it can be wind or hail. Only for the (extended) winter season, we can be relatively sure that the damage is caused by wind and not by hail. We will reformulated this part.

5. Line 127, There should be literature support here to justify the choice of 0.01%.

- Thank you for this point! We motivate the choice of this threshold in the revised manuscript (l141 -144).

6. Line 130, The symbols in the text are inconsistent with those in the formulas.

- Thanks! We changed it!

7. Line 155, I don't quite understand the meaning of "binary information." Does it mean that we can only determine whether they occurred in the same winter season?

- Good point! That is not clearly formulated. We introduce a dichotomous (binary) variable which can take take only two values, either "same season" or "previous season". We clarified this in the revised document.

8. Line 160, I also feel confused about "one seventh of all previous events fall in this time span".

- Thanks! There is a problem with the sentence. We fixed this in the revision.

9. Lines 179 –184, A significance analysis is needed here.

- This part describes the results shown in Fig. 3 which shows the estimates of the expected loss ratio as a function of previous event loss ratio conditional on various wind loads. The point to be taken from this plot is that over the range of previous events' loss ratios, the actual events' loss ratio changed. The shaded areas depict 95% confidence for the estimated of the expected loss ratios. If confidence intervals for two different previous events' loss ratios do not overlap, we can safely consider the difference as statistically significant. In the revised version, we added text to clarify this and extended the significance analysis.

10. The labels and legends in Figure 9 and Figure 10 are not clear.

- Thanks! In the revised version, we adjusted these to become clearer.

---

## Author Response (AR2)

***Response to Referee Tristian Stolte***

Thank you very much for taking the time to carefully review our revised manuscript and for your valuable and constructive feedback. We are also grateful for your recognition of the improvements made during the previous revision process.

*Introduction*

- *L32: Although the definition of vulnerability now comes at a more timely moment in the introduction than in the first manuscript, it does come a bit out of the blue (i.e., the reader may ask why hazard and exposure are not defined?). I would suggest to put a bridge in between the definition of risk and the definition of vulnerability where the authors explain that this research focusses on vulnerability.*
  - Thank you for pointing this out. We have adjusted it accordingly.

- *L65: Similar to the previous point, the remark about Drakes & Tate (2022) and Simpson et al. (2021) feels a bit out of place. I would suggest the authors to rephrase this as: "Although Simpson et al. (2021) and Drakes & Tate (2022) discuss hazard dynamics for social vulnerability, this has not been included – to our knowledge – in physical vulnerability assessments so far."*
  - Thanks, we have incorporated the comment as suggested.

*Methods*

- *L246: This sentence is hard to follow and possibly incomplete: "Using a daily scale even, if the pre-event and the event are more than one season apart lead to problems in the model approach, because of the gap between two winter seasons"*
  - Yes, you're right. We have revised it to improve clarity.

- *L237: In a comment from the previous round I mentioned 'spatial' resolution and scale where I should have written 'temporal' (as the authors rightly pointed out). Regardless, the revised text makes it much clearer and easier to follow.*
  - Thank you for addressing this once again.

-
*Results*

- *For conciseness, the authors could consider to make one plot with subplots from Figures 7-9 and one for Figures 10-12, reducing the number of plots overall.*
  - We agree with your opinion and have made the corresponding adjustments. In the first round of reviews, the second referee requested that the plots be made more readable, so we have split them into individual plots. We hope that we have now found a good balance between size, clarity, and readability.

*Discussion/Conclusion*
*Although the discussion and conclusion are now more coherent and with a more logical flow of information than in the previous manuscript, I would still like to point out some possibilities for improvement:*

- *The discussion is mostly very critical on the work, which takes away from the main message and the (seemingly) large amount of work that has been done. The points mentioned are*

*worth mentioning, but lack a positive counter part in the form of the strengths of the approach.*

- Thank you for acknowledging the strengths of this paper and the approach. We have made efforts to emphasize these more clearly.

- *This may be a style preference, but the hypotheses that follow from the results are currently presented in the conclusion, whereas this may sit better in the discussion. This also helps balance the currently negative connotation in the discussion towards a more neutral perspective on the work. For instance, I would expect a section like the following in the discussion: "Our results showed that vulnerability reduced with increasing pre-event loss ratios. We hypothesize that this happens because XXX. Looking at the data, we should also consider that this result can be influence by YYY. Given the model that we used, it is likely that ZZZ."*
  - This is an important point. Thank you for your observation. We have removed part of the conclusion/summary, have integrated it into a new subsection within the discussion and included parts of your remarks

- *The authors, in their hypotheses following the results, focus mainly on reconstruction timing. Although this is a valid point to consider, I was also wondering if the loss ratio can be affected by a loss in damageable structures? For instance, a window that is broken, or a tile that is blown off cannot be damaged again before it is repaired. Given the data and modelling approach, would the authors say that this is a point worth mentioning in the discussion/conclusion?*
  - This is an important point. You are correct that broken windows or fallen roof tiles cannot break a second time, which would reduce the vulnerability. However, damages within the house, such as those caused by rain entering through broken windows or similar, are more likely. We have incorporated this into the discussion. Thank you for the valuable suggestion.